# Learning Distributions Generated by Single-Layer ReLU Networks in the Presence of Arbitrary Outliers

**Saikiran Bulusu**[*]
Department of EECS
Syracuse University
Syracuse, NY 13244
sabulusu@syr.edu

**Geethu Joseph**[*]
Faculty of EEMCS
Delft University of Technology
2628 XE Delft, The Netherlands
g.joseph@tudelft.nl

**M. Cenk Gursoy**
Department of EECS
Syracuse University
Syracuse, NY 13244
mcgursoy@syr.edu

**Pramod K. Varshney**
Department of EECS
Syracuse University
Syracuse, NY 13244
varshney@syr.edu [*]

## Abstract

We consider a set of data samples such that a fraction of the samples are arbitrary outliers, and the rest are the output samples of a single-layer neural network with rectified linear unit (ReLU) activation. Our goal is to estimate the parameters (weight matrix and bias vector) of the neural network, assuming the bias vector to be non-negative. We estimate the network parameters using the gradient descent algorithm combined with either the median- or trimmed mean-based filters to mitigate the effect of the arbitrary outliers. We then prove that $\tilde{O}\left(\frac{1}{p^2} + \frac{1}{\epsilon^2 p}\right)$ samples and $\tilde{O}\left(\frac{d^2}{p^2} + \frac{d^2}{\epsilon^2 p}\right)$ time are sufficient for our algorithm to estimate the neural network parameters within an error of $\epsilon$ when the outlier probability is $1 - p$, where $2/3 < p \leq 1$ and the problem dimension is $d$ (with log factors being ignored here). Our theoretical and simulation results provide insights into the training complexity of ReLU neural networks in terms of the probability of outliers and problem dimension.

## 1 Introduction

Probability densities based on generative models are used in various applications such as image inpainting, deblurring, and generating new images. In particular, generative models representing the distribution over natural images are used for image inpainting and to generate new images Van Oord et al. (2016). A critical obstacle in generative modeling is building complex and expressive models that are tractable and scalable. Hence, a fundamental challenge in machine learning, as well as in statistics, is to estimate a high dimensional distribution using a set of observed data samples generated by the distribution. One solution technique is the deep generative method that models the unknown distribution as the output distribution of a neural network when the input of the neural network is drawn from a known distribution (for example, standard Gaussian in image generation by generative adversarial networks (GANs)) (Goodfellow et al., 2014; Arjovsky et al., 2017; Radford et al., 2015; Kingma, Welling, 2013; Van Oord et al., 2016). Here, the unknown distribution is estimated by learning the neural network parameters from the data samples. Several deep generative

---

[*]First two authors have equal contribution.

36th Conference on Neural Information Processing Systems (NeurIPS 2022), New Orleans, USA.

models like GANs (Goodfellow et al., 2014; Arjovsky et al., 2017; Radford et al., 2015), variational autoencoders (Kingma, Welling, 2013), and autoregressive models (Van Oord et al., 2016) have been proposed. However, the guarantees associated with such models are generally unknown. So, we address the problem of computing the sample complexity for the parameter estimation of a single-layer neural network with rectified linear unit (ReLU) activation using a corrupted dataset. Here, the corrupted dataset refers to the model wherein the output samples consists of a fraction of arbitrary outliers introduced by an adversary (Byzantine). Motivated by the deep generative models, we consider the unsupervised learning framework, i.e., we assume the knowledge of the output samples, but the corresponding inputs are unknown and drawn from the standard Gaussian distribution.

We start with a brief review of the related literature. The algorithms for neural network parameter estimation have been widely studied under both supervised and unsupervised learning frameworks. The estimation algorithms generally rely on the stochastic gradient descent (SGD)-based algorithm (Goel et al., 2018; Allen-Zhu et al., 2019; Chen et al., 2022; Oymak, 2019; Lei et al., 2020); or the gradient descent (GD)-based approach (Cao, Gu, 2019; Du et al., 2019). Further, some works have considered the sample complexity of a single-layer NN with ReLU activation in the unsupervised learning framework (Mazumdar, Rawat, 2018; Wu et al., 2019). It is known that when the bias vector is assumed to be a random vector, the column space of the weight matrix can be recovered within an error of $O(d)$ when the output dimension is $d$ (Mazumdar, Rawat, 2018). Further, when the bias vector is assumed to be nonnegative, $\tilde{O}(1/\epsilon^2)$ samples and $\tilde{O}(d^2/\epsilon^2)$ iterations are sufficient to estimate the parameters of the ReLU network within an error of $\epsilon$. However, none of the above works have considered the problem of corrupted samples. To the best of our knowledge, the estimation of a neural network in the presence of noise or outliers has been addressed only in the context of supervised learning (Bakshi et al., 2019; Goel et al., 2019; Mukherjee, Muthukumar, 2020; Zhang et al., 2019; Frei et al., 2020; Vempala, Wilmes, 2019). Our paper focuses on a special case of unsupervised learning with corrupted samples wherein we consider a single layer ReLU network.

Our neural network parameter estimation is also related to the area of robust statistics. Mathematically, the problem of learning the distribution using a neural network is equivalent to estimating the parameters of a truncated Gaussian distribution (see Section 2 for details). Robust statistics also deals with the estimation of high dimensional distributions like Gaussian, Gaussian product, and Gaussian mixture distributions where a fraction of the samples are arbitrary outliers (Huber, 2004; Hampel et al., 2011; Lai et al., 2016; Diakonikolas et al., 2019; Diakonikolas, Kane, 2019; Kane, 2021). However, parameter estimation from truncated Gaussian samples in the noisy setting has not been studied in the robust statistics literature, and we address this literature gap.

We present our major contributions as the following.

- *Learning Algorithm:* We devise a learning algorithm (Algorithm 1) to estimate the parameters (weight matrix and bias vector) of a single-layer ReLU neural network in the unsupervised learning framework where each output sample can potentially be an arbitrary outlier with a fixed probability. Our estimation algorithm uses GD along with either the median- or trimmed mean-based filters to mitigate the effect of outliers.

- *Sample Complexity:* Our algorithm requires $\tilde{O}\left(\frac{1}{p}\left[\frac{1}{p} + \frac{1}{\epsilon^2}\right]\log\frac{d}{\delta}\right)$ samples to estimate the network parameters within an error of $\epsilon$ with probability $1 - \delta$ when the probability of a sample being uncorrupted is $p \in (2/3, 1]$ and the output dimension is $d$ (Theorem 1). We also characterize the total variation distance between the estimated and true distributions. (Corollary 1).

- *Lower bound:* We derive a lower bound on the sample complexity which says that at least $\Omega(1/p\epsilon^2)$ samples are required to estimate the parameters up to an error of $\epsilon$. (Theorem 2).

- *Empirical validation (Section 5):* We empirically evaluate the performance of our algorithm and show that it is robust to the arbitrary outliers. Also, we see that the performance of our algorithm improves with the the probability of a sample being uncorrupted and the number of samples, and it increases slowly as the network output dimension grows. These observations from the empirical results are consistent with our theoretical results.

Overall, the results obtained in this paper show that parameter estimation of a single layer ReLU neural network is possible using robust gradient descent even in the presence of outliers.

## 2 ReLU Neural Network Learning Problem

Let the weight matrix of the ReLU neural network be denoted by $\boldsymbol{W} \in \mathbb{R}^{d \times m}$ and the bias vector by $\boldsymbol{b} \in \mathbb{R}^d$. The input to the neural network is denoted by the latent variable $\boldsymbol{z} \in \mathbb{R}^m$. We assume that the variable $\boldsymbol{z}$ is drawn from the standard Gaussian distribution. Thus, the output of the network is the random vector $\boldsymbol{x} \in \mathbb{R}^d$ given by $\boldsymbol{x} = \mathrm{ReLU}(\boldsymbol{W}\boldsymbol{z} + \boldsymbol{b}) \sim \mathcal{D}(\boldsymbol{W}, \boldsymbol{b})$, where $\mathcal{D}(\boldsymbol{W}, \boldsymbol{b})$ denotes the distribution of $\boldsymbol{x}$. Our goal is to estimate the unknown parameters $\boldsymbol{W}$ and $\boldsymbol{b}$ of the distribution $\mathcal{D}(\boldsymbol{W}, \boldsymbol{b})$ using the knowledge of a corrupted set of data samples $\mathcal{X} = \left\{ \boldsymbol{x}^{(n)} \in \mathbb{R}^d \right\}_{n=1}^N$. Here, we assume that a data sample in $\mathcal{X}$ follows the *Huber's p-contamination model* (Huber, 1964), i.e., a sample is drawn from $\mathcal{D}(\boldsymbol{W}, \boldsymbol{b})$ with probability $p$, and it is an arbitrary outlier drawn from an unknown distribution $\mathcal{D}_{\mathrm{out}}$ with probability $1 - p$. Hence, a given sample $\boldsymbol{x} \in \mathcal{X}$ follows the distribution $\mathcal{D}_p = p\mathcal{D}(\boldsymbol{W}, \boldsymbol{b}) + (1 - p)\mathcal{D}_{\mathrm{out}}$.

We make two observations about the learning problem. Firstly, it is known that when all the samples are from the true distribution, exponentially large number of samples are required to estimate the bias $\boldsymbol{b}$, if it can take any value from the set $\mathbb{R}^d$ (Wu et al., 2019). Naturally, the requirement on the number of samples would be worse in the presence of arbitrary outliers and so, we assume $\boldsymbol{b}$ to be non-negative. Thus, we assume the following.

**Assumption 1.** *The entries of $\boldsymbol{b} \in \boldsymbol{R}^d$ are all nonnegative.*

Secondly, the weight matrix $\boldsymbol{W}$ may not be identifiable from the distribution $\mathcal{D}(\boldsymbol{W}, \boldsymbol{b})$. In particular, for any unitary matrix $\boldsymbol{Q} \in \mathbb{R}^{m \times m}$, we have $\mathcal{D}(\boldsymbol{W}, \boldsymbol{b}) = \mathcal{D}(\boldsymbol{W}\boldsymbol{Q}, \boldsymbol{b})$. Since our goal is to learn the distribution, learning either $\boldsymbol{W}$ or $\boldsymbol{W}\boldsymbol{Q}$ is sufficient. Thus, we focus on the learnability of the underlying distribution and not the learnability of the neural network parameters. Specifically, our proposed algorithm estimates $\boldsymbol{W}\boldsymbol{W}^\mathsf{T} \in \mathbb{R}^{d \times d}$ and $\boldsymbol{b} \in \mathbb{R}^d$.

We tackle these issues using a new formulation which we discuss next.

## 3 Robust Estimation Algorithm

Our algorithm is similar to that in (Wu et al., 2019) which considers the special case of $p = 1$ and uses SGD. For the general case of $p \leq 1$, we combine the estimation framework with a robust filter to estimate the parameters.

To derive the robust estimation algorithm, we first consider a true sample $\boldsymbol{x} \sim \mathcal{D}(\boldsymbol{W}, \boldsymbol{b})$ whose $i$-th element is nonzero. We have $\boldsymbol{x}_i = \mathrm{ReLU}(\boldsymbol{W}_i^\mathsf{T}\boldsymbol{z} + \boldsymbol{b}_i) \sim \mathcal{N}^+(\boldsymbol{b}_i, \|\boldsymbol{W}_i\|^2)$, where $\mathcal{N}^+(\boldsymbol{b}_i, \|\boldsymbol{W}_i\|^2)$ is the truncated normal distribution obtained by restricting the normal distribution $\mathcal{N}(\boldsymbol{b}_i, \|\boldsymbol{W}_i\|^2)$ to the set of positive real numbers. Hence, estimation of $\boldsymbol{b}_i$ and $\|\boldsymbol{W}_i\|$ is equivalent to estimating the parameters of a one-dimensional truncated normal distribution $\mathcal{N}^+(\boldsymbol{b}_i, \|\boldsymbol{W}_i\|^2)$ using positive samples. Therefore, we estimate the parameters in two steps. In the first step, we estimate the $i$-th element $\boldsymbol{b}_i$ of the bias vector and the norm $\|\boldsymbol{W}_i\|$ of the $i$-th row of the weight matrix, for $i \in [d]$. The second step is the estimation of the angle $\theta_{ij}$ between the $i$-th row $\boldsymbol{W}_i \in \mathbb{R}^m$ and the $j$-th row $\boldsymbol{W}_j \in \mathbb{R}^m$ of the matrix $\boldsymbol{W}$, for $i, j \in [d]$. Then, the $(i, j)$-th entry of the symmetric matrix $\boldsymbol{W}\boldsymbol{W}^\mathsf{T}$ is given by $\|\boldsymbol{W}_i\| \|\boldsymbol{W}_j\| \cos(\theta_{ij})$.

The first step of the algorithm estimates the parameters of the univariate distribution given by $\mathcal{N}^+(\boldsymbol{b}_i, \|\boldsymbol{W}_i\|^2)$ using the $i$-th element of the samples $\mathcal{X}_i^+ = \{\boldsymbol{x}_i : \boldsymbol{x}_i > 0, \boldsymbol{x} \in \mathcal{X}\}$ via maximum likelihood estimation (Daskalakis et al., 2018). Then, the parameter estimates are given by $\arg\min_{\mu, \sigma^2} \ell(\mu, \sigma^2)$ where $\ell(\mu, \sigma^2)$ is the expected negative log likelihood that $\boldsymbol{x}_i \sim \mathcal{N}^+(\mu, \sigma^2)$ and the expectation is with respect to the true distribution $\boldsymbol{x}_i \sim \mathcal{N}^+(\boldsymbol{b}_i, \|\boldsymbol{W}_i\|^2)$. Further, from (Daskalakis et al., 2018), $\ell(\mu, \sigma^2)$ is a convex function of $\boldsymbol{v} = \begin{bmatrix} 1/\sigma^2 & \mu/\sigma^2 \end{bmatrix} \in \mathbb{R}^2$. Thus, we solve the convex optimization problem of maximizing $\ell(\mu, \sigma^2)$ over $\boldsymbol{v}$,

$$\boldsymbol{v}^* = \underset{\boldsymbol{v} \in \mathbb{R}^2}{\arg\min}\, \ell(\boldsymbol{v}). \tag{1}$$

The optimization problem in (1) can be solved using GD or SGD which is based on the gradient of $\ell(\boldsymbol{v})$ given by the relation (Daskalakis et al., 2018): $\nabla\ell(\boldsymbol{v}) = \boldsymbol{g} - \boldsymbol{h}(\boldsymbol{v})$ where

$$\boldsymbol{g} = \mathbb{E}_{x \sim \mathcal{N}^+\left(\boldsymbol{b}_i, \|\boldsymbol{W}_i\|^2\right)} \left\{ \begin{bmatrix} x^2/2 & -x \end{bmatrix}^\mathsf{T} \right\}, \tag{2}$$

$$\boldsymbol{h}(\boldsymbol{v}) = \mathbb{E}_{y \sim \mathcal{N}^+\left(\frac{\boldsymbol{v}_2}{\boldsymbol{v}_1}, \frac{1}{\boldsymbol{v}_1}\right)} \left\{ \begin{bmatrix} \frac{y^2}{2} & -y \end{bmatrix}^\mathsf{T} \right\} = \begin{bmatrix} \frac{\sigma^2 + \mu^2}{2} + \frac{\mu\sigma\phi(-\mu/\sigma)}{2(1 - \Phi(-\mu/\sigma))} & -\mu - \frac{\sigma\phi(-\mu/\sigma)}{1 - \Phi(-\mu/\sigma)} \end{bmatrix}^\mathsf{T}, \tag{3}$$

where $\mu = \boldsymbol{v}_2/\boldsymbol{v}_1$, $\sigma^2 = 1/\boldsymbol{v}_1$, and $\phi(\cdot)$ and $\Phi(\cdot)$ denote the probability density function and the cumulative distribution function of the standard normal distribution, respectively. The relation (3) follows directly from the closed form expressions of the first and second moments of the truncated Gaussian distribution $\mathcal{N}^+\left(\frac{\boldsymbol{v}_2}{\boldsymbol{v}_1}, \frac{1}{\boldsymbol{v}_1}\right)$, which are functions of $\boldsymbol{v}$ only (Johnson et al., 1995).

We observe that $\boldsymbol{g}$ does not depend on $\boldsymbol{v}$ and only depends on the true distribution parameters $(\boldsymbol{b}_i, \|\boldsymbol{W}_i\|^2)$, whereas $\boldsymbol{h}(\boldsymbol{v})$ does not depend on the true distribution. Therefore, the estimation of $\boldsymbol{g}$, which depends on the true distribution, can use the available data samples, and $\boldsymbol{h}(\boldsymbol{v})$ can be computed in closed form using the current iterate of $\boldsymbol{v}$. In other words, we compute the component of gradient $\boldsymbol{g}$ only once in the GD or SGD algorithm because it does not change across the iterations. This observation motivates us to use the GD algorithm to estimate the parameters instead of SGD. SGD introduces large variance due to the stochasticity and lower accuracy due to the outliers in every iteration whereas GD introduces a small error that does not depend on the algorithm iterate. This key observation and use of (3) to compute the gradient is the main difference between our algorithm and the algorithm in (Wu et al., 2019), apart from the robust estimation aspect.

To estimate $\boldsymbol{g}$, we partition $\mathcal{X}_i^+$ into batches of size $N_B$ and compute the batchwise estimate $\tilde{\boldsymbol{g}}^{(b)}$,

$$\tilde{\boldsymbol{g}}^{(b)} = \frac{1}{N_B} \sum_{x \in \mathcal{X}_{i,b}^+} \begin{bmatrix} x^2/2 & -x \end{bmatrix}^\mathsf{T}, \tag{4}$$

where $\mathcal{X}_{i,b}^+ \subset \mathcal{X}_i^+$ is the $b$-th batch of samples. We then combine the batchwise estimates using a well-known filter in the robust statistics literature such as the median or trimmed mean to handle the outliers. The median filter mitigates the effect of outliers by ensuring that if more than half of the batchwise estimates lie in an interval around the true value $\boldsymbol{g}$, then their median also lies in the same interval. Similarly, the trimmed mean prunes the outliers by removing the vectors with relatively large and small values (controlled by its parameter) and computes the estimate of $\boldsymbol{g}$ as the mean of the remaining vectors. Therefore, we obtain the gradient estimate $\tilde{\boldsymbol{g}} - \boldsymbol{h}(\boldsymbol{v})$ as

$$\tilde{\boldsymbol{g}} - \boldsymbol{h}(\boldsymbol{v}) = \text{filter}\left(\tilde{\boldsymbol{g}}^{(1)}, \tilde{\boldsymbol{g}}^{(2)}, \dots, \tilde{\boldsymbol{g}}^{\left(|\mathcal{X}_i^+|/N_B\right)}\right) - \boldsymbol{h}(\boldsymbol{v}), \tag{5}$$

where the function $\text{filter}(\cdot)$ is either median or trimmed mean, and $\boldsymbol{h}(\boldsymbol{v})$ is given in (3). Using the gradient estimate, the robust GD algorithm updates the $k$-th iterate $\boldsymbol{v}(k)$ as

$$\boldsymbol{v}(k) = P\left(\boldsymbol{v}(k-1) - \gamma(k-1)\left[\tilde{\boldsymbol{g}} - \boldsymbol{h}(\boldsymbol{v}(k-1))\right]\right), \tag{6}$$

where $\gamma(k) > 0$ is the diminishing step size and $P(\cdot)$ projects the iterate into a bounded region $\mathbb{D}_r$ as

$$\mathbb{D}_r = \left\{ \boldsymbol{v} \in \mathbb{R}^2 : 1/r \leq \boldsymbol{v}_1 \leq r, 0 \leq \boldsymbol{v}_2 \leq r \right\} \tag{7}$$

$$P(\boldsymbol{v}) = \begin{bmatrix} \min\{\max\{\boldsymbol{v}_1, 1/r\}, r\} & \min\{\max\{\boldsymbol{v}_2, 0\}, r\} \end{bmatrix}. \tag{8}$$

The projection ensures that $\ell(\boldsymbol{v})$ is a strongly convex function of $\boldsymbol{v}$, and the parameter $r$ controls the strong-convexity (Daskalakis et al., 2018) (see Section 4 for more details). The robust GD algorithm is summarized in Algorithm 2. The role of $r$ is further discussed in Section 4. This completes the first step of our algorithm based on robust GD which is summarized in Algorithm 2.

Finally, using the estimates obtained via the robust GD algorithm, we estimate $\hat{\theta}_{ij}$ similar to (Wu et al., 2019) using (Williamson, Shmoys, 2011, Lemma 6.7). Specifically, we have

$$\hat{\theta}_{ij} = \pi - 2\pi \left[ \frac{1}{N} \sum_{n=1}^{N} \mathbb{1}\left(\boldsymbol{x}_i^{(n)} > \hat{\boldsymbol{b}}_i\right) \mathbb{1}\left(\boldsymbol{x}_j^{(n)} > \hat{\boldsymbol{b}}_j\right) \right], \tag{9}$$

where $\mathbb{1}(\cdot)$ is the indicator function and $\hat{\boldsymbol{b}}$ is the output of the robust GD algorithm. The overall distribution learning algorithm is given in Algorithm 1 where $\hat{\Sigma}$ denotes the estimate of $\boldsymbol{W}\boldsymbol{W}^\mathsf{T}$.

| **Algorithm 1:** ReLU network estimation | **Algorithm 2:** Robust GD |
|---|---|
| **Input:** Samples $\mathcal{X} = \left\{ \boldsymbol{x}^{(n)} \in \mathbb{R}^d \right\}_{n=1}^{N}$ | **Input:** Positive samples $\mathcal{X}^+ \subset \mathbb{R}^+$ |
| | **Parameters:** Iterations $K$, step size $\gamma(k)$, projection parameter $r$, batch size $N_B$ |
| **1 for** $i \in [d]$ **do** | **1** $B \leftarrow |\mathcal{X}^+|/N_B$ |
| **2** $\quad \mathcal{X}_i^+ \leftarrow \{\boldsymbol{x}_i : \boldsymbol{x} \in \mathcal{X} \text{ and } \boldsymbol{x}_i > 0\}$ | **2** Compute $\{\tilde{\boldsymbol{g}}^{(b)}\}_{b=1}^{B}$ from $\mathcal{X}^+$ using (4) |
| **3** $\quad$ Compute $\hat{\boldsymbol{v}}$ using Algorithm 2 with input as $\mathcal{X}_i^+$ | **3** $\tilde{\boldsymbol{g}} \leftarrow \text{filter}\left(\tilde{\boldsymbol{g}}^{(1)}, \tilde{\boldsymbol{g}}^{(2)}, \ldots, \tilde{\boldsymbol{g}}^{(B)}\right)$ |
| **4** $\quad \hat{\boldsymbol{\Sigma}}_{i,i} \leftarrow 1/\hat{\boldsymbol{v}}_1$ | **4** $\boldsymbol{v}(0) \leftarrow \boldsymbol{0}$ |
| **5** $\quad \hat{\boldsymbol{b}}_i \leftarrow \max\{0, \hat{\boldsymbol{v}}_2/\hat{\boldsymbol{v}}_1\}$ | **5 for** $k = 1, 2, \ldots, K$ **do** |
| **6 for** $i < j \in [d]$ **do** | **6** $\quad \mu \leftarrow \frac{\boldsymbol{v}_2(k-1)}{\boldsymbol{v}_1(k-1)}; \sigma^2 \leftarrow \frac{1}{\boldsymbol{v}_1(k-1)}$ |
| **7** $\quad$ Compute $\hat{\theta}_{ij}$ using (9) | **7** $\quad$ Compute $\boldsymbol{h}(\boldsymbol{v}(k-1))$ using (3) |
| **8** $\quad \hat{\boldsymbol{\Sigma}}_{i,j} \leftarrow \sqrt{\hat{\boldsymbol{\Sigma}}_{i,i}\hat{\boldsymbol{\Sigma}}_{j,j}} \cos(\hat{\theta}_{ij})$ | **8** $\quad$ Update $\boldsymbol{v}(k)$ using (6) |
| **9** $\quad \hat{\boldsymbol{\Sigma}}_{j,i} \leftarrow \hat{\boldsymbol{\Sigma}}_{i,j}$ | |
| **Output:** $\hat{\boldsymbol{\Sigma}} \in \mathbb{R}^{d \times d}$ and $\hat{\boldsymbol{b}} \in \mathbb{R}^d$ | **Output:** $\boldsymbol{v}(K) \in \mathbb{R}^2$ |

## 4 Error Bounds for Parameter Estimation

This section provides our main result which characterizes the sample complexity for robust estimation of neural network parameters. The analysis of distributed learning in the presence of arbitrary outliers is studied in Chen et al. (2017); Yin et al. (2018). They assume that the available samples are split into a fixed number of batches and a constant fraction ($<1/2$) of batches are outliers. However, our setting assumes a probabilistic model where each data sample can be an outlier with probability $1 - p$ and there is no deterministic upper bound on the number of outliers. Consequently, each batch can have a mixture of true samples and outliers, and it is critical to choose the right batch size $N_B$ (see Proposition 2 and its discussion for details).

We rely on two propositions to arrive at the main results. The propositions establish the properties of the objective function of the optimization problem in (1) and the error bounds for the robust GD algorithm presented in Algorithm 2.

**Proposition 1.** *There exist positive constants $L$ and $\eta$ that depend only on the projection parameter $r \geq 1$ of Algorithm 2 such that the objective function $\ell(\boldsymbol{v})$ in (1) is an $\eta$−strongly convex and $L$−smooth function of $\boldsymbol{v}$ when $\boldsymbol{v} \in \mathbb{D}_r$ defined in (7). Here, $L$ is an increasing function of $r$ whereas $\eta$ is a decreasing function of $r$.*

Proposition 1 proves that the projection parameter $r$ controls the strong-convexity and smoothness parameters of the objective function. If $r$ takes a large value, it leads to a small strong-convexity parameter $\eta$ and a large smoothness parameter $L$. We use this property to interpret the role of parameter $r$ in the algorithm performance using the next result presenting the error bounds of the robust GD algorithm. To this end, we make the following assumptions to derive the error bounds:

**Assumption 2.** *The projection parameter $r \geq 1$ is such that the minimizer of (1), $\boldsymbol{v}^* \in \mathbb{D}_r$ where $\mathbb{D}_r$ is defined in (7).*

**Assumption 3.** *The step size of the robust GD algorithm in Algorithm 2 is fixed across the iterations, i.e., $\gamma(k) = \gamma$, for $k = 1, 2, \ldots, K$.*

A large value of $r$ ensures that the first assumption is satisfied. However, if any prior knowledge about the true parameters is known, the parameter $r$ can accordingly take smaller values. The second assumption is a guideline on how to choose the step size of the robust GD algorithm for the analysis. Under the above assumptions, the error bound for the robust GD algorithm is as follows.

**Proposition 2.** *Consider the robust GD algorithm in Algorithm 2 based on the median filter, which solves (1) with input as $\mathcal{X}^+$. Let $p^+ \in (1/2, 1]$ be the probability that a given sample in $\mathcal{X}^+$ follows the true distribution $\mathcal{N}^+ (\boldsymbol{v}_2^*/\boldsymbol{v}_1^*, 1/\boldsymbol{v}_1^*)$. Assume that there exist $\epsilon \in (0, 1)$, $\delta \in (0, 1/2)$, and $\zeta \in (0, 1 - 2\delta)$ such that the batch size $N_B$ satisfies*

$$N_B = \tilde{O}\left(\frac{1}{\epsilon^2}\log\frac{1}{\delta}\right) \ and \ N_B \leq \frac{1}{\log(1/p^+)}\log\frac{2(1-\delta)}{1+\zeta}. \tag{10}$$

*Then, under Assumptions 1 to 3, the output $\boldsymbol{v}(K)$ of our algorithm satisfies $\|\boldsymbol{v}(K) - \boldsymbol{v}^*\| \leq \epsilon$, with probability at least $1 - \delta$ if $|\mathcal{X}^+| = \Omega\left(\frac{N_B}{\zeta^2}\log\frac{1}{\delta}\right)$ and $K = \Omega(\log\frac{1}{\epsilon})$. Here, all the order constants, the step size $\gamma = 1/L$ in Assumption 3, and the linear convergence rate $\frac{L}{\eta+L} < 1$ depend only on the projection parameter $r$. Also, $\eta, L > 0$ are defined in Proposition 1.*

The above result indicates the role and suitable choices of the parameters: $K$, $\gamma(k)$, $r$, and $N_B$ as discussed next. The result states the number of iterations $K$ scales logarithmically with the inverse of the error $\epsilon$. Also, Assumption 3 shows that the result holds when the step size is the same across all the iterations. Finally, $r$ should be large enough to satisfy Assumption 2. However, a large $r$ leads to slower convergence because the rate of convergence is an increasing function of $r$. Finally, the algorithm gives an upper and lower bound on the batch size $N_B$. We note that for GD, the estimation error depends on the error in the first term of the gradient in (5), which is estimated using the batchwise gradient estimate in (4). The error in the batchwise gradient estimate is contributed by the outliers and the finite sample error (the difference between the sample moments computed using a finite number of samples from a distribution and the true moment of the distribution). With large batch size, the number of batches without any outliers is also small. Since the outliers are drawn from an arbitrary distribution $\mathcal{D}_{\text{out}}$, even if a batch contains one outlier, the error in the batchwise gradient estimate can be large. This observation explains the upper bound on the batch size which depends on $p^+$. It is important to note that when there are no outliers (i.e., $p^+ = 1$), there is no upper bound on the batch size. Similarly, if the batch size is small, the batchwise gradient computed using batches without outliers incurs a large finite sample error. This observation intuitively explains the lower bound on the batch size which is independent of $p^+$. In short, the upper and lower bounds on $N_B$ balances the tradeoff between the error due to the outliers and the finite sample error. Further, we note that the upper and lower bounds can be simultaneously achieved by choosing $\epsilon$ to be large enough.

We also note the restriction on $p^+$ which is not surprising. This is because the median-based methods work only if the number of outliers are smaller than that of the uncorrupted data samples, which naturally restricts the probability of outliers.

We next present our main theorem that discusses the overall complexity of our algorithm.

**Theorem 1.** *Consider the learning algorithm in Algorithm 1 that uses the meadian-based robust GD. Let $p \in (2/3, 1]$ be the probability that a given sample follows the true distribution $\mathcal{D}\left(\boldsymbol{W}\boldsymbol{W}^\mathsf{T}, \boldsymbol{b}\right)$. Assume that there exist $\epsilon \in (0, 1)$, $\delta \in (0, 1/2)$, and $\zeta \in (0, 1 - 2\delta)$ such that the batch size $N_B$ of Algorithm 2 satisfies*

$$N_B = \tilde{O}\left(\frac{1}{\epsilon^2}\log\frac{1}{\delta}\right) \text{ and } N_B \leq \frac{1}{\log(2/p)}\log\frac{2(1-\delta)}{1+\zeta}. \tag{11}$$

*Then, under Assumptions 1 to 3, the outputs $\hat{\boldsymbol{\Sigma}}$ and $\hat{\boldsymbol{b}}$ of Algorithm 1 satisfy*

$$\left\|\hat{\boldsymbol{\Sigma}} - \boldsymbol{W}\boldsymbol{W}^\mathsf{T}\right\| \leq [\epsilon + (1-p)]\|\boldsymbol{W}\|^2 \text{ and } \left\|\hat{\boldsymbol{b}} - \boldsymbol{b}\right\| \leq \epsilon\|\boldsymbol{W}\|, \tag{12}$$

*with probability at least $1 - \delta$ if the number of samples $N = \tilde{O}\left(\frac{1}{p}\left[\frac{1}{p} + \frac{1}{\zeta^2\epsilon^2}\right]\log\frac{d}{\delta}\right)$. The algorithm runs in time $\tilde{O}\left(\frac{d^2}{p}\left[\frac{1}{p} + \frac{1}{\zeta^2\epsilon^2}\right]\log\frac{d}{\delta}\right)$ and space $\tilde{O}\left(\frac{d}{p}\left[\frac{1}{p} + \frac{1}{\zeta^2\epsilon^2}\right]\log\frac{d}{\delta} + d^2\right)$. All the order constants and the step size $\gamma$ in Assumption 3 depend only on the algorithm parameter $r$.*

With no outliers ($p = 1$), our result is comparable with the existing error bounds from (Wu et al., 2019, Theorem 1). Specifically, for SGD in (Wu et al., 2019) with no outliers, the number of samples $N = \tilde{O}\left(\frac{1}{\epsilon^2}\log\frac{d}{\delta}\right)$ is sufficient to achieve

$$\|\hat{\boldsymbol{\Sigma}} - \boldsymbol{W}\boldsymbol{W}^\mathsf{T}\| \leq \epsilon\|\boldsymbol{W}\|^2 \text{ and } \|\hat{\boldsymbol{b}} - \boldsymbol{b}\| \leq \epsilon\|\boldsymbol{W}\|. \tag{13}$$

with probability at least $1 - \delta$ for any $\epsilon, \delta \in (0, 1)$, In our case, when $p = 1$, there is no upper bound on $N_B$ and we choose $N_B = N = \Omega\left(\frac{1}{\epsilon^2}\log^3\frac{d}{\delta}\right)$ to achieve (13) with probability $1 - \delta$. Thus, the time and space complexities of our algorithm are identical to those of SGD in (Wu et al., 2019). We next bound the total variation distance between the estimated distribution and the true distribution under the restriction that $\boldsymbol{W}$ is a full-rank square matrix.

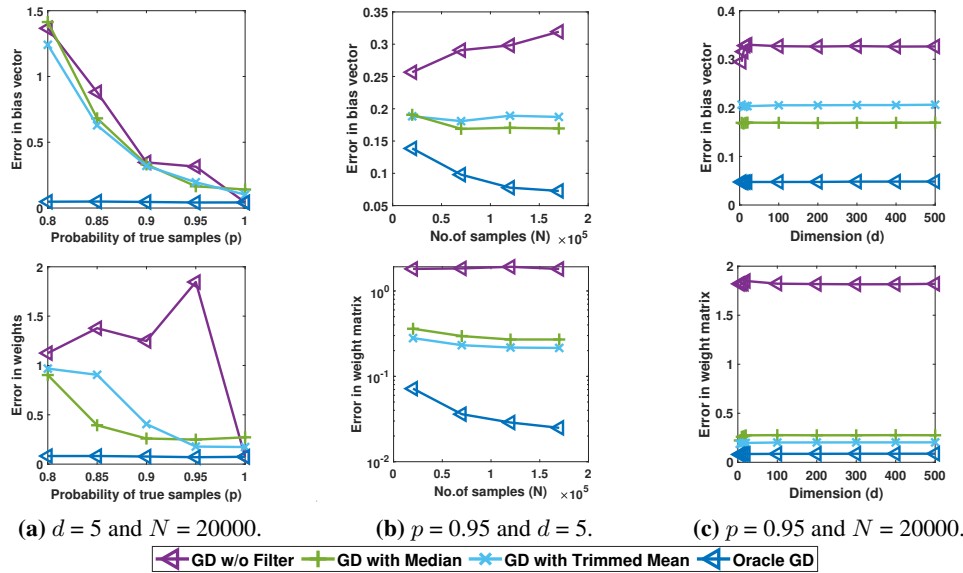

**(a)** $d = 5$ and $N = 20000$.      **(b)** $p = 0.95$ and $d = 5$.      **(c)** $p = 0.95$ and $N = 20000$.

GD w/o Filter    GD with Median    GD with Trimmed Mean    Oracle GD

**Figure 1:** Comparison of the different GD schemes as a function of $p$ (first column), $N$ (second column), and $d$ (third column). The figures indicate that utilizing the filters improves the performance of GD algorithm for mixture of samples.

**Corollary 1.** *Consider the learning algorithm in Algorithm 1 that uses the meadian-based robust GD. Suppose that $\boldsymbol{W} \in \mathbb{R}^{d \times d}$ is full-rank with $d > 1$ and let $\kappa$ be the condition number of $\boldsymbol{W}\boldsymbol{W}^{\mathsf{T}}$. Let $p > 1 - \frac{1}{2\kappa d}$ be the probability that a given sample follows the true distribution $\mathcal{D}\left(\boldsymbol{W}\boldsymbol{W}^{\mathsf{T}}, \boldsymbol{b}\right)$. Assume that there exist $\epsilon \in (\kappa d(1-p), 1/2]$, $\delta \in (0, 1/2)$, and $\zeta \in (0, 1 - 2\delta)$ such that the batch size $N_B$ of Algorithm 2 satisfies $N_B = \tilde{O}\left(\frac{\kappa^2 d^2}{(\epsilon - \kappa d(1-p))^2} \log \frac{1}{\delta}\right)$ and $N_B \leq \frac{1}{\log(2/p)} \log \frac{2(1-\delta)}{1+\zeta}$. Then, under Assumptions 1 to 3, the outputs $\hat{\boldsymbol{\Sigma}}$ and $\hat{\boldsymbol{b}}$ of Algorithm 1 satisfy*

$$\mathrm{TV}\left(\mathcal{D}\left(\hat{\boldsymbol{\Sigma}}^{1/2}, \hat{\boldsymbol{b}}\right), \mathcal{D}\left(\boldsymbol{W}, \boldsymbol{b}\right)\right) \leq \epsilon, \tag{14}$$

*with probability at least $1 - \delta$ if the number of samples $N = \tilde{O}\left(\frac{1}{p}\left[\frac{1}{p} + \frac{\kappa^2 d^2}{\zeta^2(\epsilon - \kappa d(1-p))^2}\right] \log \frac{d}{\delta}\right)$. Here, $\mathrm{TV}(\cdot)$ denotes the total variation distance between the argument distributions. All the order constants and the step size $\gamma$ in Assumption 3 depend only on the algorithm parameter $r$.*

We note that when $d > 1$, we have $1 - \frac{1}{2\kappa d} \geq 1/4$ and the bound on $p$ in Theorem 1 is automatically satisfied.

The last result of this section gives a lower bound on sample complexity of the problem of learning ReLU NN described in Section 2. We restrict the analysis to a specific class of ReLU distributions where $\boldsymbol{W}$ is a scaled identity matrix.

**Theorem 2.** *Consider the ReLU parameter estimation problem with $p$ as the probability that a given sample follows the true distribution. Suppose that the true distribution belongs to $\mathcal{C} = \{\mathcal{D}(\boldsymbol{W}, \boldsymbol{b}) : \boldsymbol{W} = \sigma \boldsymbol{I}, \boldsymbol{b} \in \mathbb{R}^d, \boldsymbol{b}_i > 0 \forall i\}$, where $\sigma = O(1)$. Then, any algorithm that learns $\mathcal{C}$ to satisfy $\left\|\hat{\boldsymbol{b}} - \boldsymbol{b}\right\| \leq \epsilon \|\boldsymbol{W}\|$ with success probability at least $2/3$ requires $\Omega\left(\frac{1}{p\epsilon^2}\right)$ samples.*

Comparing the sample complexity achieved by our algorithm (Theorem 1) and the above lower bound, we can see that the second term of our sample complexity matches the derived bound up to log factors. However, there is a gap between the sample complexity of our algorithm and the lower bound due to the first term that varies as $1/p^2$ (ignoring the log factors). This is an interesting direction for future work to see if there are better bounds.

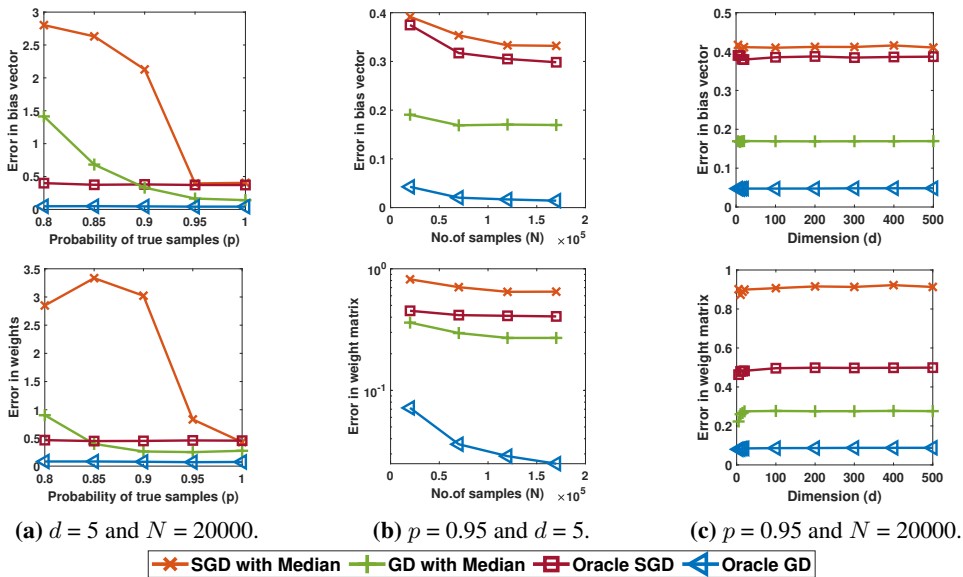

**Figure 2:** Comparison of GD and SGD schemes as a function of $p$ (first column), $N$ (second column), and $d$ (third column). We infer that SGD is more sensitive to corrupted samples compared to GD.

**Table 1:** Runtime of various schemes when $p = 0.95$, $N = 20000$, and $d = 5$. The table indicates that SGD schemes are faster than their GD counterparts.

| Scheme | Oracle | Without Filter | With Median | With Trimmed Mean |
|--------|--------|----------------|-------------|-------------------|
| GD | 16.95 s | 17.73 s | 34.44 s | 60.78 s |
| SGD | 1.24 s | 1.60 s | 2.11 s | 3.62 s |

## 5    Simulation Results

In this section, we provide numerical results to verify the performance of our algorithm. In our simulation setup, the columns of $W$ are chosen as the left singular vectors of random matrices from the standard Gaussian distribution. For $b$, we use a random vector from the standard normal distribution whose negative values are replaced with zeros. The mixture of samples are generated such that a sample comes from $\mathcal{D}(W, b)$ with probability $p$ and from $\mathcal{D}_{\text{out}}$ with probability $1 - p$. The outlier distribution $\mathcal{D}_{\text{out}} = \mathcal{N}(5, 1)$ and the algorithm hyper-parameters are $r = 3$ and $\gamma(k) = \frac{1}{0.1k}$. We use the batch-splitting approach to compute the gradient, inducing randomization. Also, from our experiments, we observe that the errors flatten certain number of iterations (see Figures 3 and 4 in the supplementary material) around $1/100$-th of the number of positive output samples which is chosen as the number of GD and SGD iterations $K$. We compute two error metrics from the estimated parameters and the ground truth, $\|\hat{\Sigma} - WW^{\mathsf{T}}\|_F / \|W\|_F^2$ and $\|\hat{b} - b\|_2 / \|W\|_F$. Further, we compare our algorithm with two other schemes: the oracle schemes (estimation using the true samples only) and schemes without a filter. Our results are shown in Figures 1 and 2, and Table 1, and the observations from them are as follows.

*Effect of the filters:* From Figure 2, the GD schemes perform better than the corresponding SGD schemes. Also from Figures 1 and 2, we infer that using filters along with GD or SGD reduces the effect of the outliers, and the curves are closer to the oracle schemes. We also infer that the median-based approach performs slightly better than the trimmed mean-based approach.

*Dependence on the probability of a sample being uncorrupted $p$:* From Figures 1a and 2a, the performance of all the schemes except the oracle schemes improves with $p$ because the fraction of outliers in the observed samples decreases with increasing $p$. However, the schemes without filters show a considerable difference in performance as they are not able to handle the outliers effectively. The performance of oracle schemes does not change with $p$ as they assume the knowledge of true samples. Further, all the schemes converge to the corresponding oracle schemes when $p = 1$.

*Dependence on the number of samples $N$ and dimension $d$:* Figures 1b and 2b show that the estimation performance of the oracle schemes and the schemes with the filter improves with the number of samples $N$. However, the schemes without a filter do not always improve with $N$ because the number of outliers also increases with $N$, which are not handled by the algorithm. In Figures 1c and 2c, we varied the dimensions up to 500 and observed that there is a slight increase in the errors as the dimension increases for our proposed schemes as well as oracle schemes. We also observe that the errors increase as $d$ increases for GD without filter due to the presence of arbitrary outliers.

*Comparison of runtimes:* From Table 1, the SGD schemes run faster as SGD utilizes only one sample for the gradient, whereas GD utilizes all the samples. The filter-based schemes have higher computation times than the ones without filters, but the runtimes of the trimmed mean-based schemes are significantly higher than those of the median-based schemes.

Overall, the median-based GD algorithm is the most effective approach to estimating the NN parameters in the presence of the outliers and is faster than the trimmed mean-based GD algorithm. Also, the median-based scheme is parameter-free and enjoys solid theoretical guarantees.

## 6  Conclusion

In this paper, we proposed an algorithm for the estimation of the parameters of a single-layer ReLU neural network from the truncated Gaussian samples where each sample was assumed to be an arbitrary outlier with a fixed probability. Our only assumption was that the bias vector was non-negative. We analyzed the sample and time complexities of the GD-based estimation algorithm combined with median-based filters to handle the outliers. The efficacy of our approach was also demonstrated using numerical experiments. Removing the non-negativity assumption on the bias vector and extending our results to multi-layer neural networks are two promising directions for future work.

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
