# OpenReview forum: "Learning Distributions Generated by Single-Layer ReLU Networks in the Presence of Arbitrary Outliers"
_NeurIPS.cc/2022/Conference — NeurIPS 2022 Accept_

### Official Review · Reviewer_cvUF · 2022-07-11

**Rating:** 5
**Confidence:** 4
**Soundness:** 3 good
**Presentation:** 3 good
**Contribution:** 2 fair

**Summary:**

The authors study the problem of learning the distribution of a generative model in the form of a one hidden-layer ReLU in the presence of arbitrary outliers (according to Huber's contamination model). The problem without outliers has been studied by Wu et al. 2019. Authors here show that $\tilde{O}(1/p^2+(1/\varepsilon^2p))$ samples and $\tilde{O}(d^2/p^2+(d^2/\varepsilon^2p))$ time is sufficient to estimate the NN parameters within error $\varepsilon$, where the outlier probability is 1-p.  They do this by providing a Robust GD algorithm . They also provide some experiments to back their theoretical analysis.

References:

Wu, Shanshan, Alexandros G. Dimakis, and Sujay Sanghavi. "Learning distributions generated by one-layer ReLU networks." Advances in neural information processing systems 32 (2019).

**Questions:**

Please see weaknesses mentioned above.

Minor comment: Line 242 has a missing link.

**Limitations:**

I don't see the limitations of this are addressed adequately. The authors say it is in Section 3 and 4, but I am unable to find them.

**Strengths And Weaknesses:**

Strengths:

This is an interesting model and the problem of trying to learn generative models from NNs are useful to study, especially the approach taken here in the view point of robust statistics.

Weakness:

1) Even in the problem of learning a single high dimensional untruncated Gaussian, a coordinate wise filter would give an error $O(p\sqrt{d})$. Here, the analysis heavily relies on a coordinate wise learning method that was proposed by Wu et al. 2019, which works without any outliers. I think one needs to use new techniques for this problem when dealing with it through the lens of Robust statistics, that works on all the dimensions.

2) Most of the techniques in Robust statistics literature also works for more powerful adversaries. Do these methods go through for those adversaries?

3) Is there an equivalent of Corollary 1 in Wu et al. 2019 that can be obtained here, that connects the TV distances?

4) The experiments here are performed for very low dimensions upto 20, which is not as enlightening.

5) Most of the techniques are a combination of Wu et al. 2019 and Daskalakis et al. 2018.

UPDATE:
After going through the rebuttal, I believe that the authors have responded convincingly. I still believe that this is somewhat incremental in the light of previous work Wu et al. 2019 and Daskalakis et al. 2018 and to improve the bounds one has to move beyond a coordinate wise approach, as done in Robust statistics literature and maybe look at more powerful adversarial models. But I will increase my score from 3 to 5.

References:

Daskalakis, Constantinos, et al. "Efficient statistics, in high dimensions, from truncated samples." 2018 IEEE 59th Annual Symposium on Foundations of Computer Science (FOCS). IEEE, 2018.

Wu, Shanshan, Alexandros G. Dimakis, and Sujay Sanghavi. "Learning distributions generated by one-layer ReLU networks." Advances in neural information processing systems 32 (2019).

---

> ### Author Response · Authors · 2022-08-02
> **We thank the reviewer for their comments. The reviewer was mainly concerned with the significance of the problem of learning the distribution generated by a NN in the presence of arbitrary outliers, the techniques used, and the TV distance between distributions. Specifically, We address the reviewer's concerns as below and have also made changes to the main draft.**
>
> **Q1:** *Even in the problem of learning a single high dimensional untruncated Gaussian, a coordinate-wise filter would give an error $O(p\sqrt{d})$. Here, the analysis heavily relies on a coordinate-wise learning method that was proposed by Wu et al. 2019, which works without any outliers. I think one needs to use new techniques for this problem when dealing with it through the lens of Robust statistics, which works on all dimensions.*
>
> **A1:** We agree that our analysis relies on the coordinate-wise learning method similar to that in Wu et al. However, note that learning a single high dimensional truncated Gaussian distribution is fundamentally different and more challenging than learning a Guassian distribution. So it is not clear if $O(p\sqrt{d})$ can be achieved using a vectorized approach. For example, Daskalakis et al. shows that (not coordinate-wise) estimation of a truncated Gaussian distribution without outliers leads to an error of $O(d/\sqrt{n})$. However, a tight lower bound on the error is still an open problem even for the setting with no outliers and the robust analysis is more challenging (see A5).
>
> **Q2:** *Most of the techniques in Robust statistics literature also works for more powerful adversaries. Do these methods go through for those adversaries?*
>
> **A2:** We note that in our work, the assumption is that the outliers are from an arbitrary distribution $\mathcal{D}_{\text{out}}$ with no restrictions. Hence, our setting is quite general with respect to the outliers. Specifically, we use median-of-means wherein the outliers that are too high or low in magnitude are handled by the median and the remaining ones are smoothed by the mean. Therefore, our algorithm is robust to outliers from any distribution.
>
> **Q3:** *Is there an equivalent of Corollary 1 in Wu et al. 2019 that can be obtained here, that connects the TV distances?*
>
> **A3:** We have obtained the bound for the total variation distance between the estimated distribution and the true distribution under the restriction that ${W}$ is a full-rank square matrix in Section 4 (see Corollary 1). Our result states that when $p>1-\frac{1}{2\kappa d}$, if there exist $\epsilon \in (\kappa d (1-p), 1/2]$ $\delta\in(0,1/2)$, and $\zeta\in(0, 1-2\delta)$ such that the batch size $N_B$ satisfies $N_B=\tilde{O}\left(\frac{\kappa^2d^2}{(\epsilon-\kappa d (1-p))^2}\log\frac{1}{\delta}\right)$ and $N_B \leq \frac{1}{\log (2/p)}\log \frac{2(1-\delta)}{1+\zeta}$, then the TV distance is less than $\epsilon$,  with probability at least $1-\delta$ if the number of samples $N=\tilde{O}\left( \frac{1}{p}\left[ \frac{1}{p}+\frac{\kappa^2d^2}{\zeta^2(\epsilon-\kappa d (1-p))^2}\right]\log\frac{d}{\delta}\right)$. These results reduce to Corollary 1 in Wu et al. for the special case of $p=1$.
>
> **Q4:** *The experiments here are performed for very low dimensions upto 20, which is not as enlightening.*
>
> **A4:** We added new experiments for dimensions up to 500 and observed that there is a slight increase in the errors as the dimension increases. Specifically, we did not observe major changes in the trends as the dimension increased from 20 to 500, and we expect this trend to continue for higher dimensions as well.
>
> **Q5:** *Most of the techniques are a combination of Wu et al. 2019 and Daskalakis et al. 2018.*
>
> **A5:** We agree that our techniques are influenced by Wu et al. and Daskalakis et al. We note that extending the results in Wu et al. to our probabilistic setting is not straightforward using the tools from outlier robustness analysis. In distributed robust optimization, the batch size is decided by the number of available machines, out of which a constant fraction (<1/2) is adversarial. So the median is computed over the batch gradients, of which only less than half are outliers. However, in our case, each batch is a mixture of true samples and outliers with no deterministic upper bound on the number of outliers. Also, our batch size is not predefined but a design parameter. Further, median-based GD typically makes several assumptions on the gradient, such as bounded skewness (Yin et al.) and linear growth of the moments (Yang et al.), which are not satisfied by the ReLU or truncated Gaussian distribution. Moreover, Wu et al. and Daskalakis et al. use SGD, which leads to finite sample error in each iteration, making analysis cumbersome. Instead, we used GD to compute the data sample-based component of the gradient using data samples (computed only once) and the closed-form expressions of the truncated Gaussian distribution moments for the data sample-independent component of the gradient in every iteration. This simplified the analysis since the gradient error is independent of the iterate. Finally, since the first step of the algorithm only handles positive samples, probabilities $p^+$ and $p$ were involved in the analysis and deriving a simple relation between them was challenging.
>
> **Q6:** *Line 242 has a missing link*
>
> **A6:** Fixed.

---

> > ### Author Response · Authors · 2022-08-02
> > **We include the references cited earlier in our response.**
> >
> > References:
> >
> > Wu et al. "Learning distributions generated by one-layer ReLU networks." NeurIPS, 2019.
> >
> > Daskalakis et al. "Efficient statistics, in high dimensions, from truncated samples." IEEE FOCS, 2018.
> >
> > Yin et al. "Byzantine-robust distributed learning: Towards optimal statistical rates." ICML, 2018.
> >
> > Yang et al. "Byzantine-resilient stochastic gradient descent for distributed learning: A lipschitz-inspired coordinate-wise median approach." IEEE CDC, 2019.

---

> > ### Comment · Reviewer_cvUF · 2022-08-08
> > **Thank You Authors**
> >
> > Dear Authors,
> >
> >  In light of the rebuttal I have increased my score from 3 to 5.

---

> > > ### Author Response · Authors · 2022-08-09
> > > **We thank the reviewer for raising the score and recommending borderline accept**
> > >
> > > >*UPDATE: After going through the rebuttal, I believe that the authors have responded convincingly. I still believe that this is somewhat incremental in the light of previous work Wu et al. 2019 and Daskalakis et al. 2018 and to improve the bounds one has to move beyond a coordinate-wise approach, as done in Robust statistics literature and maybe look at more powerful adversarial models. But I will increase my score from 3 to 5.*
> > >
> > > We thank the reviewer for the positive feedback and constructive criticism.
> > >
> > > We reiterate that our work generalizes Wu et al using the standard and well-studied Hubers contamination model without any restriction on the distribution of the outliers. It is not clear to us what " more powerful adversarial models" refer to.
> > >
> > > Also, in robust statistics, it is known that the (non-coordinate-wise) maximum likelihood estimation of the parameters of a truncated Gaussian distribution without outliers leads to a sample complexity of $O(d/\sqrt{n})$ and the bound is optimal (Daskalakis et al. 2018). Moreover, the coordinate-wise approach is corroborated by our new experimental results where we have added new experiments for dimensions up to 500. We observed that there is a slight increase in the errors as the dimension increases from 20 to 500. Hence, the dimension has little effect on the coordinate-wise approach, and so, moving away from the coordinate-wise approach does not seem to improve the bounds.
> > >
> > > We also note that we have further strengthened our paper by adding a new lower bound on the estimation error/sample complexity in the revised draft. The lower bound matches the sample complexity of our algorithm, and its proof does not use any techniques from Wu et al.

---

### Official Review · Reviewer_KLJj · 2022-07-12

**Rating:** 6
**Confidence:** 4
**Soundness:** 4 excellent
**Presentation:** 4 excellent
**Contribution:** 3 good

**Summary:**

The authors consider the problem of learning a one layer generative model with adversarial corruptions. A one layer generative model is a standard Gaussian input followed by a linear transformation then a ReLU activation. They consider the contamination model where a fraction of the data comes from an arbitrary outlier distribution.

They show that the gradient descent algorithm of Wu et al. can be made into a robust algorithm. Specifically, they show that you can do robust gradient descent to learn the mean and variance of each coordinate, then do robust estimation of the angle coordinates to get the full covariance matrix. They experimentally verify their algorithm and show that robust gradient descent can outperform the non-robust algorithm on corrupted data.

**Questions:**

Why is a new strong convexity proof required? I see Wu et al. mentions strong convexity and the parameter r but the discussion is somewhat different.

It is interesting that the $b$ parameter error is not affected by the outlier probability. What is the intuition for this?

**Limitations:**

I found no issues here.

**Strengths And Weaknesses:**

Strengths:

The authors establish a clean result on robust learning of 1 layer generative models with ReLU activations.

The paper is well written and easy to understand.

The authors present empirical evaluations.

Weaknesses:

The approach of Wu et al. breaks the problem down into a series of scalar optimization and estimation problems. Robust statistics is significantly easier in scalar situations, so making the algorithm of Wu et al. robust is somewhat incremental.

One way to improve the paper would be to discuss lower bounds based on the outlier fraction.

Minor Issues:

242: There is a missing reference.

The theorems are not in latex theorem blocks, which makes it hard to isolate them from the surrounding text.

---

> ### Author Response · Authors · 2022-08-02
> **We thank the reviewer for their comments. The reviewer was mainly concerned with the significance of the problem of learning the distribution generated by a NN in the presence of arbitrary outliers, the techniques used, and the lower bounds based on outlier fraction. We address the reviewer's concerns as below and have also made changes to the main draft.**
>
> **Q1:** *The approach of Wu et al. breaks the problem down into a series of scalar optimization and estimation problems. Robust statistics is significantly easier in scalar situations, so making the algorithm of Wu et al. robust is somewhat incremental.*
>
> **A1:** We agree that our approach builds on the fact that the problem can be broken down into a series of scalar optimization and estimation problems like Wu et al. However, we estimate both bias and row norm of the weight matrix using robust GD and so we do not deal with scalars but two-dimensional vectors. Nonetheless, extending the results in Wu et al. to our probabilistic setting is not straightforward using the existing tools from outlier robustness analysis because of the following challenges:
> 1. In distributed robust optimization, the batch size is decided by the number of available machines of which a constant fraction (<1/2) is adversarial. So the median is computed over the batch gradients of which only less than half are outliers. However, for us, each batch is a mixture of true samples and outliers with no deterministic upper bound on the number of outliers. Also, our batch size is not predefined but a design parameter.
> 2. Median-based GD makes several assumptions on the gradient, such as bounded skewness (Yin et al.) and linear growth of the moments (Yang et al.), which are not satisfied by the ReLU or truncated Gaussian distribution.
> 3. Wu et al. and Daskalakis et al. use SGD, which leads to finite sample error in each iteration, making analysis cumbersome. Instead, we used GD to compute the data-based component of the gradient using data samples (computed only once). This simplified the analysis since the gradient error is independent of the iterate.
> 4. Since the first step of the algorithm only handles positive samples, the analysis involves probabilities $p^+$ and $p$, and deriving a simple relation between them was challenging.
>
> We also note that as the reviewer suggested, we have added a new lower bound on the estimation error (Theorem 2) that does not use any techniques from Wu et al. or Daskalakis et al., but matches our derived sample complexity.
>
> **Q2:** *One way to improve the paper would be to discuss lower bounds based on the outlier fraction.*
>
> **A2:** We have obtained a lower bound on the sample complexity of the problem of learning the ReLU neural network in Section 4 (see Theorem 2). We show that any algorithm that learns a special class of ReLU neural networks needs  $\Omega\left(\frac{1}{p\epsilon^2}\right)$ samples to satisfy $\|\hat{{b}}-{b}\|\leq \epsilon\|{W}\|$ with success probability at least 2/3. Comparing the result with the complexity achieved by our algorithm, we observe that the second term of our sample complexity matches the derived bound up to log factors. However, there is a gap between the sample complexity of our algorithm and the lower bound due to the first term that varies as $1/p^2$ (ignoring the log factors). Moreover, we have obtained the bound for the total variation distance between the estimated distribution and the true distribution under the restriction that ${W}$ is a full-rank square matrix in Section 4 (see Corollary 1).
>
> **Q3:** *242: There is a missing reference.*
>
> **A3:** Fixed.
>
> **Q4:** *The theorems are not in latex theorem blocks, which makes it hard to isolate them from the surrounding text.*
>
> **A4:** Fixed.
>
> **Q5:** *Why is a new strong convexity proof required? I see Wu et al. mentions strong convexity and the parameter r but the discussion is somewhat different.*
>
> **A5:** We agree that Wu et al. mentions the strong convexity property of the objective function and also the parameter $r$. However, the role of $r$ and its choice is not explicitly discussed in Wu et al, which is explained in the discussions following Propositions 1 and 2. Also, our proof relies on the existing results (Lemma 1 and 2) and only mentions the connection between strong convexity, smoothness and $r$.
>
> **Q6:** *It is interesting that the $\mathbf{b}$ parameter error is not affected by the outlier probability. What is the intuition for this?*
>
> **A6:** The outlier probability affects the error in bias estimation. The error bound $\epsilon$ affects the lower bound on $N_B$ and the sample complexity, which in turn depends on the outlier probability. For example, as the outlier probability $1-p$ increases, the number of samples required to achieve the same error $\epsilon$ increases. In other words, for a fixed sample complexity, the error increases with the outlier probability, which is intuitive. A similar dependence is also observed via the lower bound on $N_B$. Further, the errors in the bias and weight matrix are slightly different and the error in weights depends on both $\epsilon$ and $p$. This is because the error in weights comes from both row norm and angle estimations (the first and second steps of the algorithm) whereas the error in the bias is only from the first step of the algorithm.

---

> > ### Author Response · Authors · 2022-08-02
> > **We include the references cited earlier in our response.**
> >
> > References:
> >
> > Wu et al. "Learning distributions generated by one-layer ReLU networks." NeurIPS, 2019.
> >
> > Daskalakis et al. "Efficient statistics, in high dimensions, from truncated samples." IEEE FOCS, 2018.
> >
> > Yin et al. "Byzantine-robust distributed learning: Towards optimal statistical rates." ICML, 2018.
> >
> > Yang et al. "Byzantine-resilient stochastic gradient descent for distributed learning: A lipschitz-inspired coordinate-wise median approach." IEEE CDC, 2019.

---

> > > ### Comment · Reviewer_KLJj · 2022-08-09
> > > **Thank you for your response**
> > >
> > > Thank you for your response showing the novelty of the work. I will raise my score from a 5 to a 6.

---

### Official Review · Reviewer_eN7y · 2022-07-15

**Rating:** 6
**Confidence:** 4
**Soundness:** 3 good
**Presentation:** 3 good
**Contribution:** 2 fair

**Summary:**

This paper proposes an algorithm to learn the distribution generated by single layer ReLU generative models. Prior work of Wu et al showed that an algorithmic adaptation of Daskalakis et al can be used to learn the weights (upto rotation) and biases of the neural network.

This work shows that a similar algorithm can be used to learn the network parameters, even in the presence of outliers under Huber's contamination model. The authors propose an algorithm that combines Wu et al with robust estimation algorithms such as median-of-means and trimmed mean estimators.

**Questions:**

- I do not understand how this algorithm can possibly work for all $0 \le p < 1$. It's easy to see that $p < 0.5$, then this would imply that the majority of batches are corrupted and hence the median estimate will also be corrupted. This can also be verified in the Appendix, Line 456, where the inequality cannot be satisfied for any $N_B$ if $p^+$ is smaller than 0.5. In fact, the bound in line 456 places a strong upper bound on $N_B$ and I think that satisfying this along with the lower bound may not be as trivial as the authors claim.

- Please comment on the non-independence of the gradients after the first iteration.

**Strengths And Weaknesses:**

Strengths:
- The algorithm is intuitive and easy to understand, and the presentation is mostly clear. The proofs seem mostly correct and the evaluation shows that it is competitive with Wu et al for $p=1$ and is better for $p < 1$ (as expected).

Weaknesses:
- The algorithm feels like a merge of median-of-means with the SGD approach of Wu et al and Daskalakis et al, and the authors should clarify what are the novel technical challenges they faced in the proof of the main Theorem.

- This algorithm cannot work if $p \le 0.5$, in which case the majority of batches are corrupted, and the median estimate is not useful. This is not mentioned in the paper, and the authors claim it works for all $ 0 < p \le 1$.

- After the first gradient step, the per-sample gradients computed in later batches are no longer independent of one another, and the concentration bounds used are no longer valid. I think this can be fixed assuming a fresh batch of samples is received at each time step, and a union bound should give the failure probability as $ K \delta $. But the authors should be more careful and include this dependence in their bounds. I think this also affects the number of batches needed, but I'm not sure.

---

> ### Author Response · Authors · 2022-08-02
> **We thank the reviewer for their comments. The reviewer was mainly concerned with the significance of the problem of learning the distribution generated by a NN in the presence of arbitrary outliers, the techniques used, and the independence of the gradients after the first iteration. Specifically, We address the reviewer's concerns as below and have also made changes to the main draft.**
>
> **Q1:** *Algorithm feels like a merge of median-of-means with the SGD approach of Wu et al and Daskalakis et al, and the authors should clarify what are the novel technical challenges they faced in the proof of the main Theorem.*
>
> **A1:** Outlier robustness analysis of Wu et al. using the existing tools is nontrivial due to the following challenges:
> 1. In distributed robust optimization, the batch size is decided by the number of available machines of which a constant fraction (<1/2) is adversarial. So the median is computed over the batch gradients of which only less than half are outliers. However, each of our batch is a mixture of true samples and outliers with no deterministic upper bound on the number of outliers. Also, our batch size is not predefined but a design parameter.
> 2. Median-based GD makes several assumptions on the gradient, such as bounded skewness (Yin et al.) and linear growth of the moments (Yang et al.), which are not satisfied by ReLU or truncated Gaussian distribution.
> 3. Wu et al. and Daskalakis et al. use SGD, which leads to finite sample error in each iteration, making analysis cumbersome. Instead, we used GD to compute the data-based component of the gradient using data samples (computed only once). This simplified the analysis since the gradient error is independent of the iterate.
> 4. Since the first step of the algorithm only handles positive samples, the analysis involves probabilities $p^+$ and $p$, and deriving a simple relation between them was challenging.
>
> We also note that we have added a new lower bound on the estimation error (Theorem 2) that does not use any techniques from Wu et al. or Daskalakis et al., but matches our derived sample complexity.
>
> **Q2:** *Algorithm cannot work if $p\leq 0.5$, in which case the majority of batches are corrupted, and the median estimate is not useful. This is not mentioned in the paper, and the authors claim it works for all $0\leq p\leq 1$.*
>
> **A2:** As the reviewer has rightly pointed out, the analysis does not hold if $p^+\leq 1/2$. We have changed the proof of Theorem 1 to show that $p^+\geq p/(2-p)$. Hence, $p>2/3$ is sufficient for the lower bound on $p^+$ to hold. We have corrected the manuscript accordingly.
>
> **Q3:** *After the first gradient step, the per-sample gradients computed in later batches are no longer independent of one another, and the concentration bounds used are no longer valid. I think this can be fixed assuming a fresh batch of samples is received at each time step, and a union bound should give the failure probability as $K\delta$. But the authors should be more careful and include this dependence in their bounds. I think this also affects the number of batches needed, but I'm not sure.*
>
> **A3:**  We note that from (2) and (3), the estimated gradient has two components: (i) $\tilde{\mathbf{g}}$ which does not depend on $\mathbf{v}$ and only depends on the true distribution parameters $(\mathbf{b}_i,\|\mathbf{W}_i\|^2)$, and (ii) $\mathbf{h(v)}$ which only depends upon $\mathbf{v}$ and does not depend on the true distribution. Further, $\tilde{\mathbf{g}}$ that depends on the true distribution is computed from the available data samples, and $\mathbf{h(v)}$ is computed in closed form using the current iterate of $\mathbf{v}$. Hence, we need to compute the component of the gradient that depends on data samples only once because it does not change across the iterations. This key observation and use of $\mathbf{h(v)}$ to compute the gradient is the main difference between our approach and Wu et al., apart from robust estimation. Furthermore, in the analysis, the concentration bounds are applied only once on $\|\tilde{\mathbf{g}}-{\mathbf{g}}\|$ as the computation of $\mathbf{h(v)}$ is exact. So union bound or change in the number of batches is not needed. We have modified Section 3 to highlight this point.
>
> **Q4:** *I do not understand how this algorithm can possibly work for all $0\leq p<1$. It's easy to see that $p<0.5$, then this would imply that the majority of batches are corrupted and hence the median estimate will also be corrupted. This can also be verified in the Appendix, Line 456, where the inequality cannot be satisfied for any $N_B$ if $p^+$ is smaller than 0.5. In fact, the bound in line 456 places a strong upper bound on $N_B$ and I think that satisfying this along with the lower bound may not be as trivial as the authors claim.*
>
> **A4:** We have changed the proof of Theorem 1 to show that $p>2/3$ is a sufficient condition for our results. Further, the lower bound on $N_B$ in Proposition 1 and Theorem 1 depends on $1/\epsilon^2$, which can be achieved by choosing $\epsilon$ to be large enough. We have added this point to the paper. Similarly, in Corollary 1, the lower bound on $N_B$ holds for large values of $\epsilon$ and $p$ for which $\epsilon/(\kappa d)- (1-p)$ is large enough.
>
> **Q5:** *Please comment on the non-independence of the gradients after the first iteration.*
>
> **A5:** See A3.

---

> > ### Author Response · Authors · 2022-08-02
> > **We include the references cited earlier in our response.**
> >
> > References:
> >
> > Wu et al. "Learning distributions generated by one-layer ReLU networks." NeurIPS, 2019.
> >
> > Daskalakis et al. "Efficient statistics, in high dimensions, from truncated samples." IEEE FOCS, 2018.
> >
> > Yin et al. "Byzantine-robust distributed learning: Towards optimal statistical rates." ICML, 2018.
> >
> > Yang et al. "Byzantine-resilient stochastic gradient descent for distributed learning: A lipschitz-inspired coordinate-wise median approach." IEEE CDC, 2019.

---

### Meta-Review · Area_Chair_rcKG · 2022-08-30

**Recommendation:** Accept
**Confidence:** Certain

**Metareview:**

This paper learns the distributions created by single layer ReLU generative models. The paper extends previous work on this problem in the presence of outliers under Huber's contamination model.

The key innovation of the new algorithm is the use of robust estimation and its combination with the previous method to robustify it. The authors argued about the technical issues that arise in robustifying the work of Wu et al. and did a good job in addressing all the reviewer comments.

This paper is novel, well written and well motivated.
Unfortunately the problem setting is still narrow because the authors can only learn single layer generative models, but still this turns out to be non-trivial. The proposed solution involves solid technical contributions that were well explained.


**Award:**

No

---

### Decision · Program_Chairs · 2022-09-14

Accept